# Freezing suppression by oxytocin in central amygdala allows alternate defensive behaviours and mother-pup interactions

Elizabeth Rickenbacher[1], Rosemarie E Perry[2,3,4,5,6], Regina M Sullivan[4,5,6], Marta A Moita[1]*

[1]Champalimaud Neuroscience Programme, LIsboa, Portugal; [2]Department of Neuroscience and Physiology, New York University School of Medicine, New York, United States; [3]Sackler Institute of Graduate Biomedical Sciences, New York University School of Medicine, New York, United States; [4]Emotional Brain Institute, Nathan Kline Institute for Psychiatric Research, New York, United States; [5]New York University Child Study Center, New York University School of Medicine, New York, United States; [6]Department of Child and Adolescent Psychiatry, New York University School of Medicine, New York, United States

*For correspondence: marta.moita@neuro.fchampalimaud.org

Competing interests: The authors declare that no competing interests exist.

**Abstract** When animals and their offspring are threatened, parents switch from self-defense to offspring protection. How self-defense is suppressed remains elusive. We postulated that suppression of the self-defense response, freezing, is gated via oxytocin acting in the centro-lateral amygdala (CeL). We found that rat dams conditioned to fear an odor, froze when tested alone, whereas if pups were present, they remained in close contact with them or targeted the threat. Furthermore, blocking oxytocin signaling in the CeL prevented the suppression of maternal freezing. Finally, pups exposed to the odor in the presence of the conditioned dam later froze when re-exposed alone. However, if oxytocin signaling in the dam had been blocked, pups failed to learn. This study provides a functional role for the well-described action of oxytocin in the central amygdala, and demonstrates that self-defense suppression allows for active pup protection and mother-pup interactions crucial for pup threat learning.

## Introduction

There are several examples in the wild of active defense responses by parents to threats that constitute a danger not only to the offspring but also themselves (*Byrkjedal, 1989*; *Pavel and Bureš, 2001*; *Long, 1993*). Rather than engaging in the self-defense behavior of hiding or fleeing, the parents might hide the young and/or attack a predator. Parental defense is wide spread across taxa, including fish (*Carlisle, 1985*), reptiles (*Greene et al., 2006*), birds (*Ghalambor and Martin, 2001*) and mammals (*Blank et al., 2015*), strongly suggesting that it has been selected multiple times. Despite its clear adaptive value, little is known about the neural mechanisms by which parents flexibly change from self-defense to offspring defense. To address this issue we used the laboratory rat, for which a great deal is known regarding the neural circuits underlying defense behaviors. Evidence that rat dams suppress their own defense responses when threatened in the presence of pups is scarce (*Pinel et al., 1990*; *Sukikara et al., 2010*). However, it is well described that the social environment modulates defensive behavior, such as freezing in adult rats (*Kikusui et al., 2006*; *Kiyokawa et al., 2004 Knapska et al., 2010*) and that rat dams will attack an adult male that poses

**eLife digest** Animals have many mechanisms to avoid or defend themselves against deadly encounters with predators. However, adult animals frequently put themselves at risk while protecting their more vulnerable offspring from attacks. For example, a killdeerbird with young will fake a broken wing and lead a predator away from its nest. This helps ensure that the parent's genes live on and contribute to the survival of their species. To do this, the parent must override his or her own defense mechanisms and protect the young instead of themselves.

Little is known about the exact mechanisms that allow animals to suppress their own defense mechanisms while protecting their young. Freezing is one tactic that animals will use when they are unable to escape a predator. Previously, studies have shown that the hormone oxytocin, which is produced in the brain, suppresses freezing behavior. Oxytocin plays an important role in birth and breastfeeding, but it is also known to strengthen the bond between individuals, in particular between mother and child. Until now, it was not known whether this hormone also blocks self-defense behaviors in animals protecting their offspring.

Now, Rickenbacher et al. show that oxytocin does indeed block freezing behavior, enabling mother rats to protect their offspring in the face of a threatening smell. In the experiments, mother rats were taught to fear the scent of peppermint. Without their young, these rats would freeze whenever they smelled peppermint. Yet, when mother rats with their pups were exposed to the scent, they did not freeze. Instead, they tried to defend their young. Blocking oxytocin in a part of the mothers' brains called the amygdala, however, caused them to freeze in response to the scent of peppermint, even in the presence of their pups.

The experiments show that oxytocin helps mother rats suppress their self-defense mechanisms and is necessary for the mothers to protect their young. Rickenbacher et al. also showed that pups of oxytocin-treated mothers did not learn to freeze in response to the threat. But pups of untreated mothers who defended them, learned to freeze when they were exposed to the scent of peppermint. A next step will be to record neurons that produce oxytocin to better understand how the presence of the pups stimulate its production in their mothers. In addition, it is still unclear how pups learn from their mothers to freeze in response to a threat. One possibility is that the mother produces a molecule that signals danger. Identifying this molecule would be the next step.

a danger to the pups but not themselves, whereas virgin females do not (*Bosch, 2013*; *Giovenardi et al., 2000*; *Knapska et al., 2010*; *Mayer and Rosenblatt, 1987*).

Oxytocin (OT), a highly conserved neuropeptide, has been implicated in a number of maternal behaviors including nursing, pup retrieval and maternal aggression (*Carter, 2014*; *Insel et al., 1998*; *Pedersen et al., 1992*) involving brain regions such as the medial preoptic area, anteroventral peri-ventricular nucleus of the hypothalamus, auditory cortex and lateral septum among others (*Bosch et al., 2004*; *Leng et al., 2008*; *Marlin et al., 2015*; *de Almeida et al., 2014*; *Scott et al., 2015*). In addition, oxytocin suppresses freezing, a conserved defense response to inescapable threats (*Bale et al., 2001*; *Huber et al., 2005*; *Knobloch et al., 2012*; *Veenema and Neumann, 2008*; *Viviani et al., 2011*; *Windle et al., 1997*; *Zoicas et al., 2014*). In rodents, oxytocinergic neu-rons from the peri-ventricular hypothalamus (PVN) project to the centro-lateral nucleus of the amyg-dala (CeL) (*Knobloch et al., 2012*), a brain region that regulates conditioned threat responses (*Fanselow and Wassum, 2015*; *Haubensak et al., 2010*; *Ciocchi et al., 2010*; *Wilensky et al., 2006*; *Li et al., 2013*; *Ciocchi et al., 2010*). In this amygdaloid sub-nucleus the release of oxytocin excites CeL-off cells thereby inhibiting freezing to a learned threat (*Bosch and Neumann, 2012*; *Huber et al., 2005*; *Knobloch et al., 2012*; *Viviani et al., 2010*). Although the mechanism by which oxytocin regulates freezing is characterized in great detail, the natural circumstances under which animals rely on it remains untested.

We hypothesized that oxytocin in the CeL is required for the suppression of freezing in mothers, thereby permitting active protection of their offspring when both mothers and pups are threatened. To test our hypothesis, we established a paradigm to assess the modulation of maternal defensive responses by the presence of their pups. Thereafter, we used targeted injections of an oxytocin

antagonist (OTA) to the CeL, and tested the role of this neuropeptide in the modulation of maternal defense. Our findings highlight a functional role for oxytocin in the central amygdala, by indicating that oxytocin in the CeL suppresses maternal freezing allowing active pup protection in the face of threat, and maternal transmission of fear to the pups present during the threat. Furthermore, our findings provide novel evidence for the modulation of active maternal defense responses, based on the age of pups present during threat.

## Results

### Regulation of dams' defensive behavior by the presence of pups

We conditioned mothers to freeze to a peppermint odor by pairing it with footshocks, and then exposed them to the conditioned odor either alone or in the presence of their pups. Rat dams were first habituated to the test chamber with their pups and the next day placed in the conditioning chamber alone, wherein they received three odor-shock pairings. On the third day, defensive behaviors of the dams in response to the conditioned odor was tested. Testing consisted of exposure of dams, either alone or with their pups, to three presentations of the conditioned odor in the test chamber, (see *Figure 1a*). From birth until weaning, many changes take place in mothers and pups. These include decreases in nursing and maternal aggression, as well as increases in the mobility of the pups (pups initially do not leave the nest and closer to weaning leave the nest frequently), and feeding behavior (pups initially are fully dependent on the mother's milk and close to weaning eat regular food chow while still nursing) (*Bolles and Woods, 1964*; *Caughey et al., 2011*; *Lonstein and Gammie, 2002*; *de Almeida et al., 2014*). We therefore tested mothers at two post-partum (PP) stages (early, PP4-6 vs. late, PP19-21).

Conditioned dams from both age groups froze when exposed to the conditioned peppermint odor alone (immobility levels increased during exposure to the odor for both age groups: PP4-6 from $6.1 \pm 3.2\%$ during baseline to $85.7 \pm 6\%$ post odor exposure; PP19-20 from $3.5 \pm 2.4\%$ during baseline to $80.6 \pm 9.3\%$ post odor exposure, See *Figure 1* and *Video 1*). Conditioned dams exposed to the peppermint odor in the presence of their pups behaved very differently, showing very little freezing behavior (PP4-6 baseline freezing 0% and freezing during odor exposure $0.7 \pm 0.5\%$; PP19-20 baseline freezing $0.1 \pm 0.1\%$ and freezing during odor exposure $0.4 \pm 0.2\%$, *Figure 1*). When comparing the increase in freezing upon odor exposure across dams tested alone versus tested with their offspring we found a significant difference (PP4-6, $n_{(w)}=8$ and $n_{(wo)}=8$, U = 0.0, p=0.000; PP19-21, $n_{(w)}=8$ and $n_{(wo)}=8$, U = 0.0, p=0.000; *Figure 1C and D* top panels).

When tested in the presence of their pups, PP4-6 dams spent much of their time pushing the bedding towards the tube that delivered the threatening peppermint odor. This behavior may correspond to defensive burying reported elsewhere (*Blanchard et al., 2005*; *De Boer and Koolhaas, 2003*; *Poling et al., 1981*). PP4-6 dams also showed increased locomotion suggestive of exploration of the nesting area. We categorized these behaviors as responses directed towards the threat: PP4-6 from $2.6 \pm 1.2\%$ during baseline to $45.2 \pm 6\%$ post odor exposure (*Figure 1E* and *Video 1*). Dams with the older pups showed very few displays of defense behaviors towards the threat: PP19-21 from $2.2 \pm 1\%$ during baseline to $3.6 \pm 1.7\%$ post odor exposure (*Figure 1F*). When comparing the increase in behaviors targeted towards the threat across dams tested alone versus tested with their offspring we found a significant difference only for dams of younger pups (PP4-6, $n_{(w)}=8$ and $n_{(wo)}=8$, U = 0.0, p=0.000 and PP19-21, $n_{(w)}=8$ and $n_{(wo)}=8$, U = 27.5, p=0.645; *Figure 1E and F*). In contrast, dams of older pups spent most of their time in close contact with the pups, either nursing or grooming them. We quantified the time spent in contact with the pups (towards pups category), and found that these dams increase the amount of time spent with their pups upon exposure to the conditioned threat (from $23.1 \pm 14.3\%$ during baseline to $72.1 \pm 5.7\%$ post odor exposure, see *Figure 1* and *Video 1*). In contrast, dams with the younger pups did not increase the amount of time spent in contact with them (from $0.2 \pm 0.2\%$ during baseline to $4.3 \pm 3\%$ post odor exposure, See *Figure 1G*). Comparing the increase in time spent with the pups upon exposure to the peppermint odor reveals a significant difference between mothers of young and older pups ($n_{(w)}=8$ and $n_{(wo)}=8$, U = 10.0, p=0.02, *Figure 1G*).

These findings clearly demonstrate that dams suppress freezing when in the presence of their pups. In addition, dams of younger and older pups behaved quite differently. Whereas the PP4-6

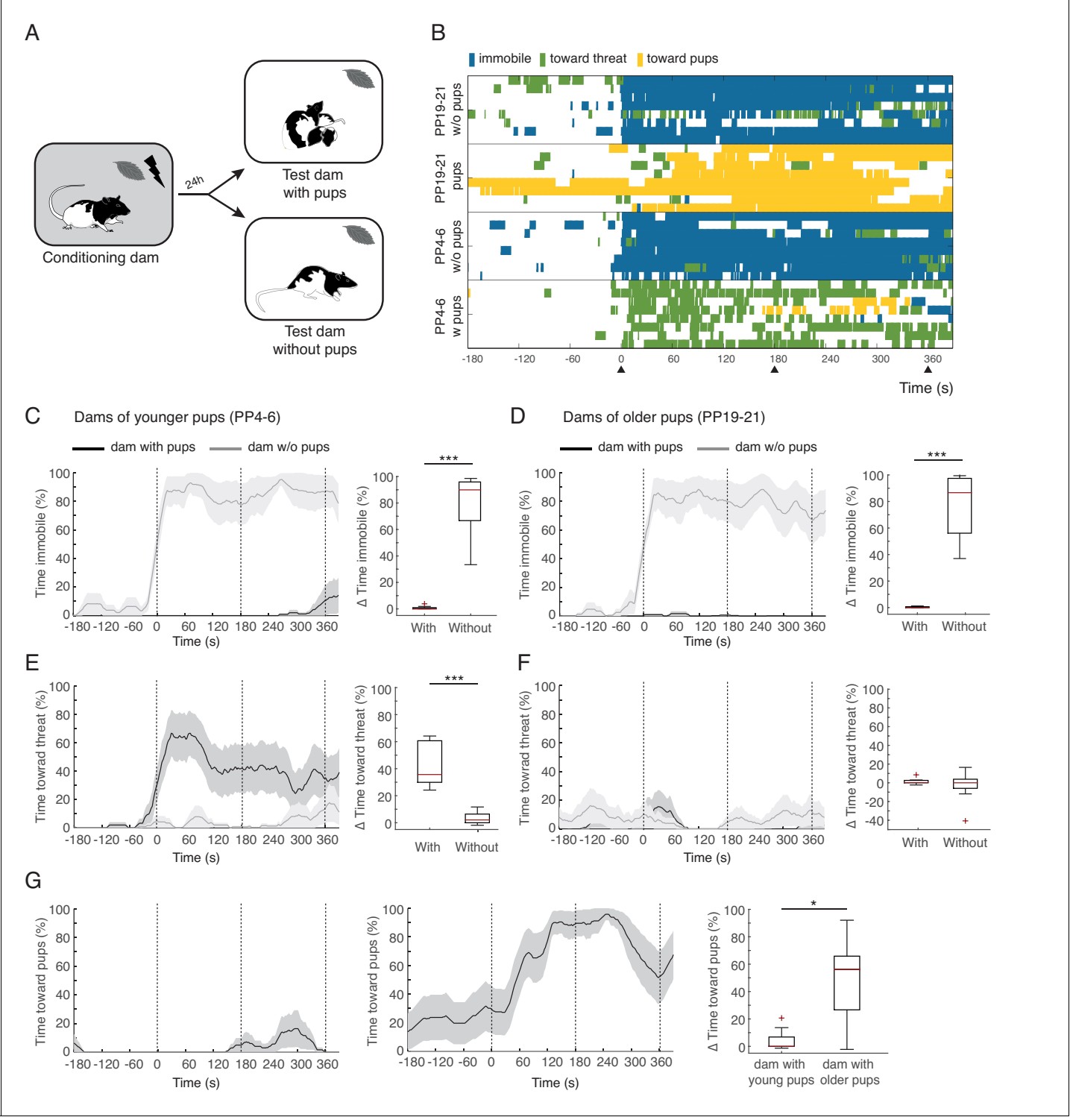

**Figure 1.** Rat dams suppress freezing when in the presence of their pups. (A) Schematic of the experimental design. (B) Raster plot showing behaviors of dams during test to the conditioned odor. Each row corresponds to one dam. Dams are sorted by treatment, PP19-21 dams with (n = 8) and without pups (n = 8), PP4-6 dams with (n = 8) and without them (n = 8). Colors correspond to the behaviors exhibited. Black triangles indicate time of odor presentations. (C) and (D) Immobility of dams with and without pups for PP4-6 dams and PP19-21 dams, respectively. Line graph shows timecourse of immobility (average ± SEM). Vertical dashed lines indicate presentation of conditioned odor. Box plot shows change in immobility upon odor presentation (immobility after first odor – baseline). (E) and (F) same as C and D for responses targeted to the threat. G Same as C-F showing time spent in contact with the pups. Box plot here compares behavior of PP4-6 dams to PP19-21 dams. * denotes p<0.05 and ***, p<0.001.

**Video 1.** Video clip taken during the test session, when conditioned dams were re-exposed to conditioned odor with and without pups (PP4-6 and PP19-21). Video clip starts at the time of first odor delivery.

dams spent most of the time directing their behavior towards the threat, the PP19-21 dams spent most of the time in close contact with the pups. This difference could be due to differences in maternal experience and/or distinct internal states of the dams. Indeed, maternal aggression, the amount of time spent nursing, and circulating hormonal levels in the dams, including those of oxytocin, are quite distinct across these stages of pup development (*Bosch and Neumann, 2012*; *Bosch, 2013*; *Cramer et al., 1990*; *Higuchi et al., 1985*; *Thiels et al., 1990*). In addition, the behavior of the pups is different, younger pups do not move away from the nest, whereas older pups have a behavioral repertoire associated with impending independent living that includes increased time outside the nest, eating solid food, relying less on nursing and expression of some defensive behaviors, including freezing (*Thiels et al., 1990*).

## Role of maternal PP stage and age of pups in determining maternal defense response

To assess whether it is the postpartum stage of the dams or the age of the pups that determines maternal defense behavior, we tested PP4-6 dams with older pups (postnatal days, PN19-23) and vice versa (PP19-23 dams with PN4-6 pups). In this case, the behavioral/physiological postpartum stage of the dam does not correspond to the behavioral/physiological age of the pups during testing. The behavior of PP4-6 dams tested in the presence of older pups (PN19-23), remained unaltered, i.e. as before they spent most of the time directing their behavior towards the threat (comparison of time sent directing behavior towards threat during the matched age test with mismatched age test: n=10, Z= −0.561, p=0.575; same comparison for behaviors towards pups: Z= −0.533, p=0.594; *Figure 2B*), suggesting that the behavior of the dams is mostly determined by the maternal stage. Conversely, PP19-21 dams, when tested with younger pups, changed their behavior such that they now targeted their response more towards the threat, decreasing the time spent with the pups (comparison of behavior during the matched age test with mismatched age test; behaviors towards threat: n=12, Z= −2.845, p=0.004, behaviors towards pups: Z= −2.824, p=0.005; *Figure 2C*), suggesting that in this case, the behavior of the dams was mostly determined by the behavior of the pups. This is in accordance with previous findings showing that maternal aggression is initially determined by the internal state of the mother but progressively becomes more dependent on the presence of the pups (*Caughey et al., 2011*; *de Almeida et al., 2014*).

## Role of endogenous oxytocin within the CeL

Next, we tested our hypothesis that activation of OT-receptors in CeL suppresses freezing, permitting expression of maternal defensive behaviors. To this end, we implanted dams bilaterally with injection cannulas into central amygdala (although we did not specifically target CeL, only this region of central amygdala expresses OT-receptors, [*Huber et al., 2005*]). After recovery from surgery, implanted dams were conditioned as before. However, in this experiment, 15 min before the test, conditioned dams received micro-infusions of either an oxytocin antagonist (OTA-dams), or PBS vehicle. In addition, we tested intact dams alongside both PBS and OTA-dams, to assess whether the infusion procedure itself did not disrupt the behavior of dams. As no differences were found between PBS and intact dams, henceforth we report the pooled data from the two groups (CONT-Dams; see Materials and methods section). Exposure to the conditioned odor was performed in the test chamber in the presence of pups. We first tested PP4-6 dams with PN4-6 pups and found that OTA-dams (n=8), but not CONT-dams (PBS infused n=6, intact n=4), displayed increased levels of freezing during odor exposure (OTA-dams from 5.2 ± 3.0% during baseline to 60 ± 8.2% post odor exposure; CONT-dams from 0% during baseline to 0.1 ± 0.1% post odor exposure; Mann-Whitney-U

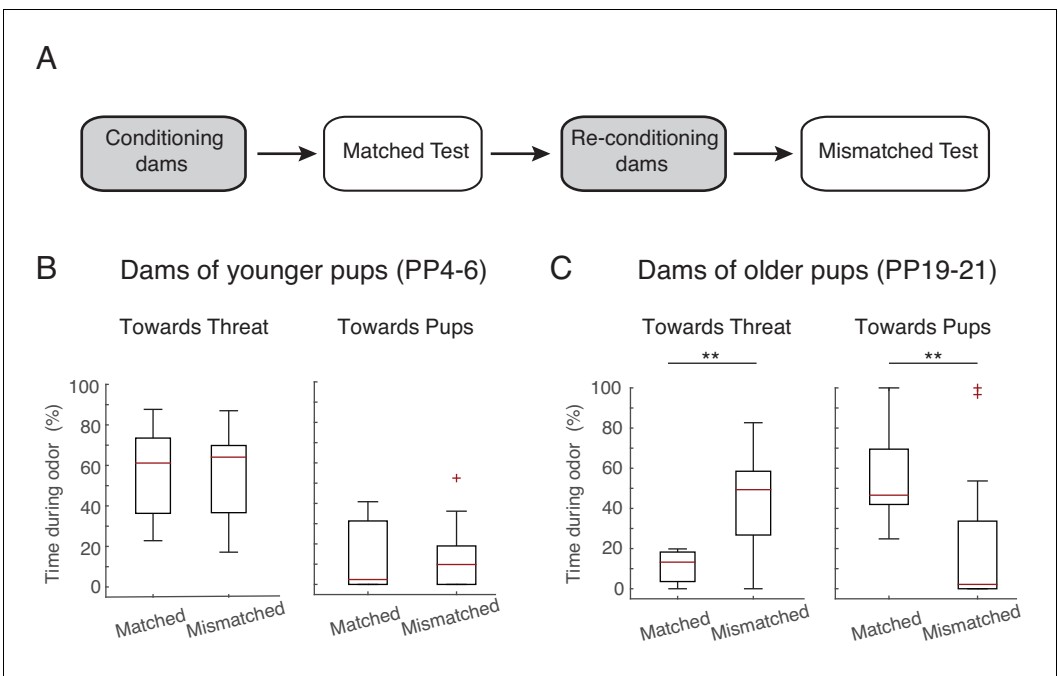

**Figure 2.** Maternal response is dependent on stage. (**A**) Schematic of experimental design. Rat dams (PP4-6, n=10, PP19-21, n=12) were first exposed with their own pups (ex. PP4-6 pups), reconditioned, and then re-exposed to the threat in the presence of age mismatched pups (PP19-21 pups). (**B**) Box plots show time PP4-6 dams spent (after 1 st odor) behaving towards the threat (left) or pups (right) when in the presence of their own (matched, PP4-6 pups) and in the presence of PP19-21 pups (mismatched). (**C**) Same as in B for PP19-21 dams. ** denotes p<0.01.

test comparing change in freezing by OTA and CONT-dams: n(CONT)=10 and n(OTA)=8, U = 0.0, p=0.000; *Figure 3*). Concomitantly, OTA infusion into CeL abolished behaviors targeting the threat or the pups (Mann-Whitney-U test comparing change in behaviors towards threat displayed by OTA and CONT-dams: n(CONT)=10 and n(OTA)=8, U=0.0, p=0.000; Mann-Whitney-U test comparing change in behaviors towards pups displayed by OTA and CONT-dams: n(CONT)=10 and n(OTA)=8, U=16.0, p=0.034; *Figure 3*).

Next, we tested PP19-21 dams with PN19-21 pups and again found that OTA-dams (n=7), but not CONT-dams (again data from PBS, n=7 and intact dams n=10 was pooled), displayed increased levels of freezing during odor exposure (OTA-dams from 0.4 ± 0.4% during baseline to 63.4 ± 5.7% post odor exposure; CONT-dams from 0% during baseline to 0% post odor exposure; Mann-Whitney-U test comparing change in freezing by OTA and CONT dams: n(CONT)=17 and n(OTA)=7, U=0.0, p=0.000; *Figure 4*). PP19-21 dams receiving OTA injections failed to increase the time spent with the pups upon odor exposure (Mann-Whitney-U test comparing change in behaviors towards pups displayed by OTA and CONT dams: n(CONT)=17 and n(OTA)=7, U = 6.0, p=0.000; *Figure 4*) and as CONT-dams did not target their actions towards the threat either (Mann-Whitney-U test comparing change in behaviors towards threat displayed by OTA and CONT dams: n(CONT)=17 and n(OTA)=7, U = 56.0, p=0.852; *Figure 4*). These results suggest that in the presence of a threat, oxytocin in the CeL suppresses freezing, allowing for the expression of pup defense. Even though a previous study disrupting oxytocin signaling in the central amygdala had no effect on maternal behaviors (*Lubin et al., 2003*), being other brain areas such as the medial preoptic area and the bed nucleus of the stria terminalis crucially implicated in maternal care (*Numan, 2006*; *Tsuneoka et al., 2013*), it is possible that in our experiments, oxytocin in the CeL contributes to normal mother-pup interactions. We addressed this issue by testing the behavior of unconditioned dams towards pups during exposure to the peppermint odor (in this case neutral). We tested PP19-20 dams, as these dams when conditioned spend most of the time during exposure to the conditioned odor in close contact with the pups. In this experiment, dams were habituated to the peppermint odor without shock such

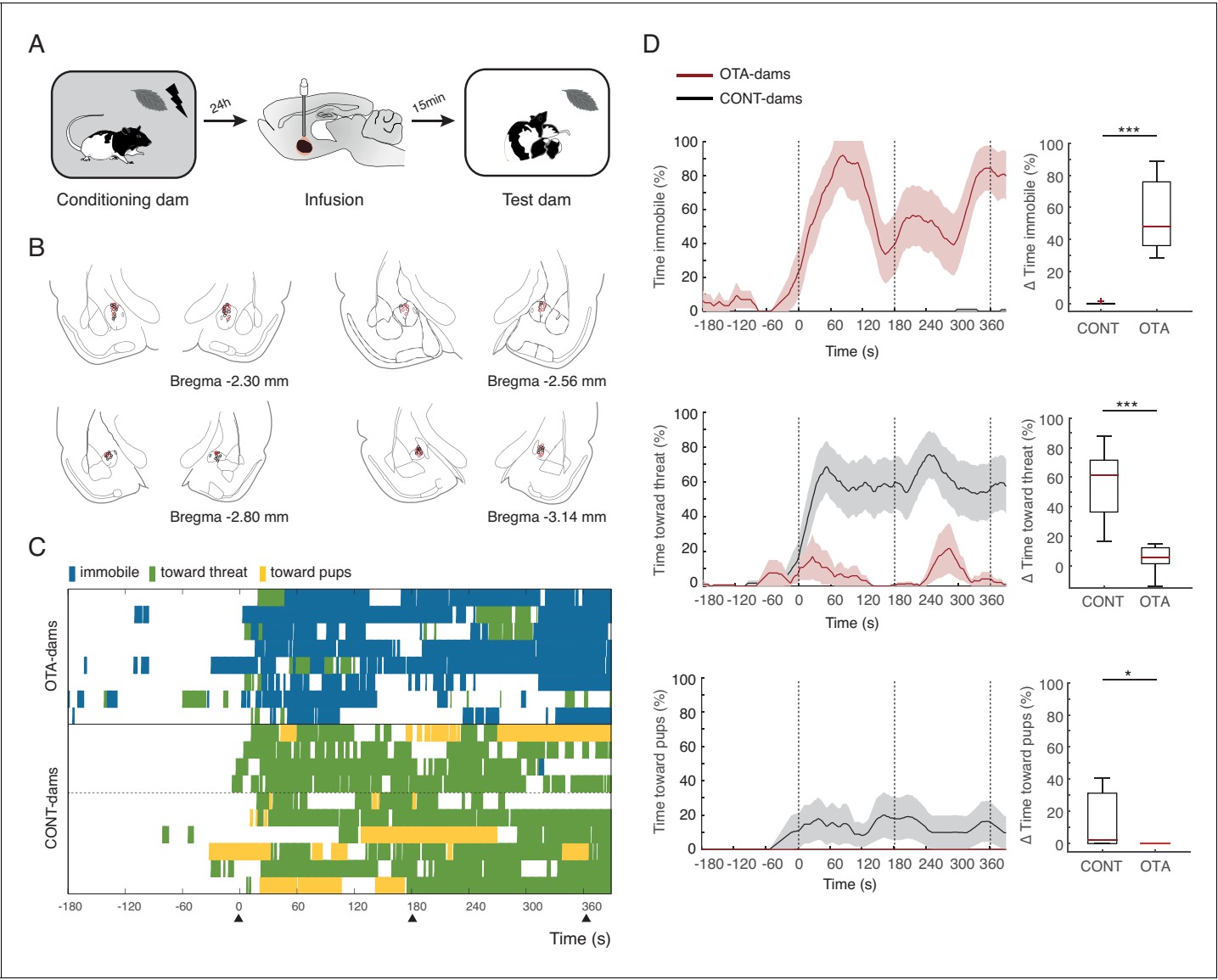

**Figure 3.** Freezing suppression by oxytocin in central amygdala in PP4-6 dams allows alternate maternal defensive behaviors and mother-pup interactions during threat. (**A**) Schematic of experimental design. (**B**) Schematic of coronal sections (adapted from Paxino and Watson's atlas) with empty black and red circles corresponding to CONT and OTA cannula placements, respectively. (**C**) Raster plot showing behaviors of PP4-6 dams during test to the conditioned odor. Each row corresponds to one dam. Dams are sorted by treatment; OXT antagonist (n = 8) or CONT (n = 10; four intact dams, above dashed line, and 6 PBS-dams, bellow dashed line). Colors correspond to behaviors exhibited. Black triangles indicate time of odor presentations. (**D**) Line graph shows timecourse of immobility (top), responses toward threat (middle) and toward pup (bottom) exhibited by either the OTA or CONT PP4-6 dams (average and ± SEM). Vertical dashed lines indicate presentation of conditioned odor. Box plots show change in these behaviors displayed by OTA and CONT dams upon odor exposure. * denotes p<0.05 and ***, p<0.001.

that the odor would not be perceived as a threat. As described above, dams were again infused with either OTA or PBS 15 min prior to the test. We found no difference in the amount of time dams spent with the pups in either group (Mann-Whitney-U test comparing change in behaviors towards pups displayed by OTA and CONT dams: $n_{(CONT)}=8$ and $n_{(OTA)}=10$, U=29.5, p=0.360), indicating that blocking oxytocin in the CeL only affects mother-pup interactions if a threat is present.

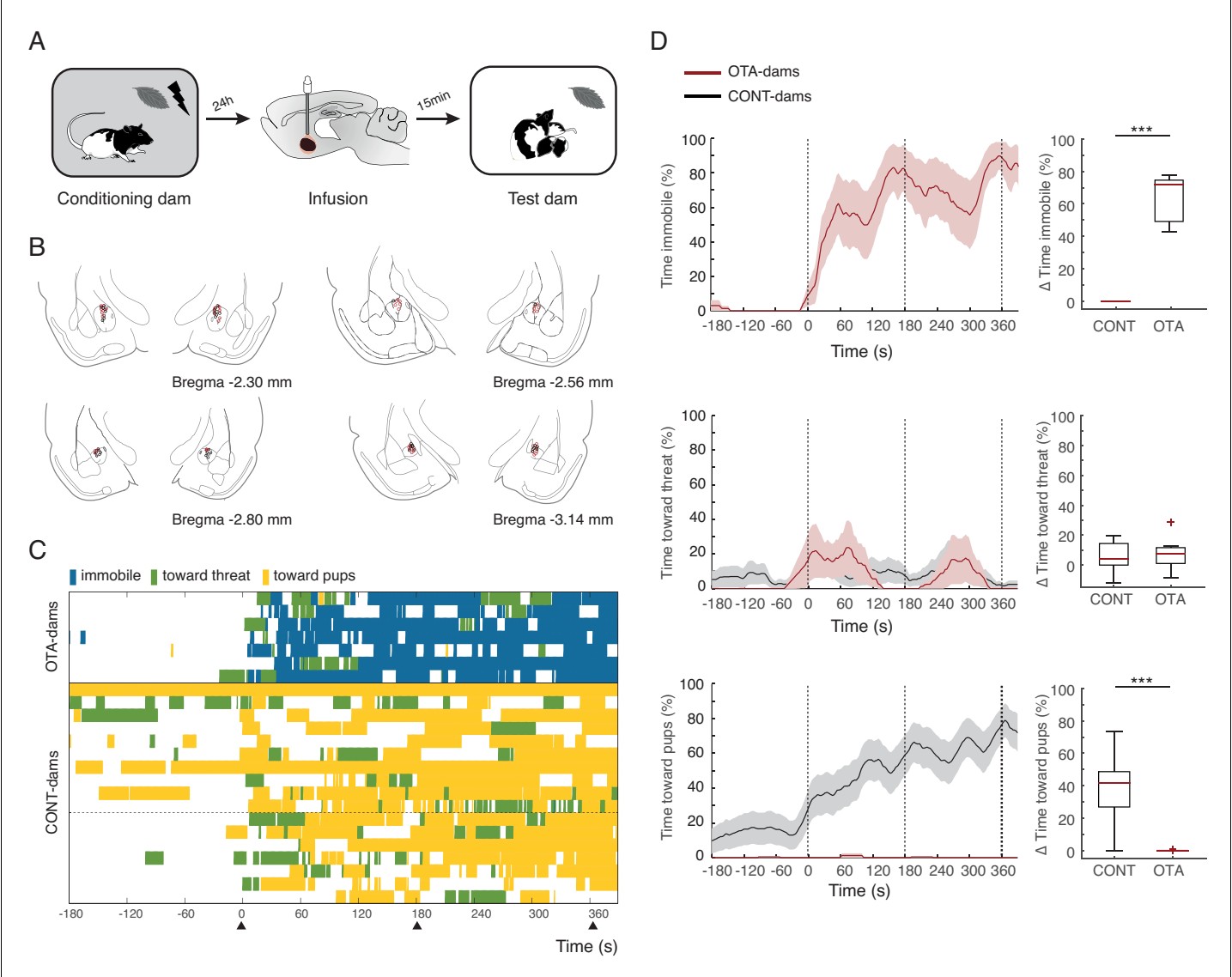

**Figure 4.** Freezing suppression by oxytocin in central amygdala in PP19-21 dams allows alternate maternal defensive behaviors and mother-pup interactions during threat. (**A**) Schematic of experimental design. (**B**) Schematic of coronal sections (adapted from Paxino and Watson's atlas) with empty black and red circles corresponding to PBS and OTA cannula placements, respectively. (**C**) Raster plot showing behaviors of PP19-21 dams during test to the conditioned odor. Each row corresponds to one dam. Dams are sorted by treatment; OXT antagonist (n = 7) or CONT (n = 17; 10 intact dams, above dashed line, and 10 PBS-dams, below dashed line; more dams belong to the control group because a subset was used for the transmission of fear from mother to pup experiment (*Figure 5*)). Colors correspond to behaviors exhibited. Black triangles indicate time of odor presentations. (**D**) Line graph shows timecourse of immobility (top), responses toward threat (middle) and toward pup (bottom) exhibited by PP19-21 dam that received OTA injections or CONT dams (average and ± SEM). Vertical dashed lines indicate presentation of conditioned odor. Box plots show change in these behaviors displayed by OTA and CONT dams upon odor exposure. \*\*\* denotes p<0.001.

## Learning from mothers about threat

It has previously been shown that young rodents learn to regulate their defensive behaviors through their mothers (*Magrath et al., 2006*; *Mateo and Holmes, 1997*; *Wiedenmayer, 2009*). Rat pups learn to avoid an odor to which they were exposed in the presence of a mother that was conditioned to respond to that odor as a threat (using a similar paradigm as the one reported here) (*Chang and Debiec, 2016*; *Debiec and Sullivan, 2014*). Therefore, we asked whether affecting normal maternal defensive behavior during exposure to the threat, by blocking oxytocin signaling in CeL of dams,

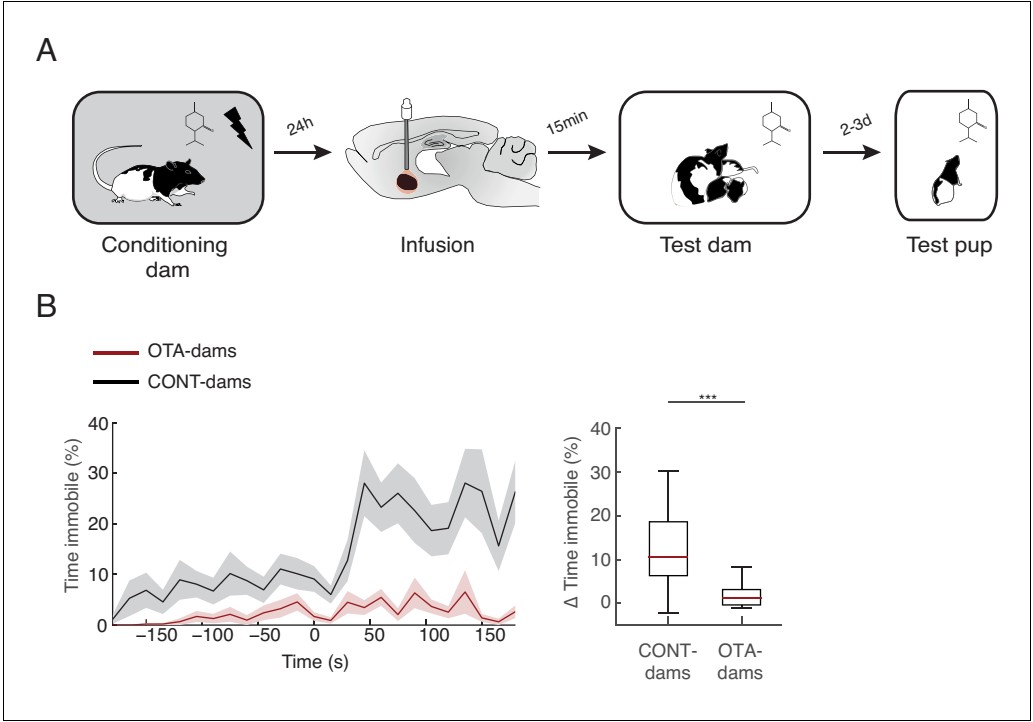

**Figure 5.** Transmission of fear from mother to pup. (**A**) Schematic of the experimental design. PP19-21 pups were re-exposed to the odor alone 2–3 days after exposure with rat dam. (**B**) Line graph shows timecourse of immobility (average ± SEM) displayed by pups (from CONT-dams, n=21, nine from intact and 12 from vehicle infused dams; from OTA-dams, n=18) during the test to the conditioned odor. Vertical dashed lines indicate presentation of conditioned odor. Box plot shows change in immobility upon odor presentation (immobility after first odor – baseline). *** denotes p<0.001.

could also affect learning about the threat by the pups. To address this issue, we tested the pups that were exposed to the peppermint odor at PN19-20 with conditioned dams that received OTA injections (therefore froze most of the time) or CONT dams (which maintained close contact with the pups). Importantly, we tested PP19-20 dams with PN19-20 pups, as learning to avoid an odor from the mother was tested with pups at slightly older age (PN6-7) then our young pup group (PN4-6) and rat pups do not learn amygdala-dependent defense behaviors in response to a self-experienced threat, such as shock, before PN10 (*Barr et al., 2009*; *Moriceau et al., 2010*; *Sullivan et al., 2000*; *Wiedenmayer and Barr, 1998*). Testing of these pups was performed 2–3 days after exposure to the peppermint odor with the conditioned dams, using a standard CS cue test. Specifically, pups were individually tested in a large glass beaker where the peppermint odor was delivered. Freezing was quantified as a measure of learning. We found that only the pups of the conditioned CONT-dams learned to freeze to the peppermint odor (pups of OTA-dams from 1.4 ± 0.3% during baseline to 3.2 ± 0.8% post odor exposure; pups of CONT-dams from 7.4 ± 1.7% during baseline to 20.2 ± 2.8% post odor exposure; Mann-Whitney-U test comparing change in freezing by pups of OTA and CONT dams: $n_{(CONT)}$=21 and $n_{(OTA)}$=18, U=52.5, p=0.000; *Figure 5*). Given the variance in the levels of freezing displayed by pups of CONT-dams, we investigated whether these were more similar within than between litters and found similar variance within and between litters (data not shown). Since OTA-dams spent most of the time freezing, whereas control dams spent more time with the pups during odor exposure, our finding suggests that close proximity between mother and pups may be required for pups to show normal learning about the threat. This is consistent with a previous study showing that a chemical cue might underlie maternal mediated threat learning by pups (*Debiec and Sullivan, 2014*). This chemical cue might constitute a short-range signal that in our testing conditions, due to the air extraction system, requires close proximity between the source, the

mother, and receiver, the pup. It is worth noting that our testing conditions may more closely reflect the natural, open-air conditions, existent in the wild.

## Discussion

In summary, the present findings provide clear evidence that rat dams modulate their defenses by the presence of offspring. When exposed to the inescapable threat alone, rat dams freeze robustly, but no freezing is observed when the pups are present. Instead, rat dams with young pups, unable to move from the nest, display defensive responses geared towards the threat actively protecting the pups. Conversely, rat dams with older and mobile pups, display increased contact with them. One possibility is that increased contact with the pups, which keeps them in close proximity, constitutes a form of defense by preventing the pups from approaching the threat. The modulation of maternal defense by pup age may be mediated by maternal and/or offspring changes. Indeed, testing mothers at a stage mismatched with that of the pups age (PP4-6 dams with PN19-21 pups and vice versa) revealed that both the age of the pups, possibly through differences in the pups behavior (*Fleming et al., 1999*), and maternal PP stage, which could be caused by differences in hormonal state and/or maternal experience (*Bosch and Neumann, 2008*; *Rosenblatt, 1967*), determine maternal defensive behavior. Furthermore, we found that maternal defense is initially determined by maternal stage progressively becoming more dependent on the pups, a similar pattern to that observed for maternal aggression (*Caughey et al., 2011*; *de Almeida et al., 2014*). Further experiments are required to parse out the mediating factors of pup age on maternal defense behaviors. Previous studies have shown social buffering of anxiety and fear in rodents (*Burkett et al., 2016*; *Hennessy et al., 2009*; *Kikusui et al., 2006*; *Waldherr and Neumann, 2007*). Particularly relevant are the studies showing that male and female adult rats freeze less to a conditioned cue when in the presence of other male rats (*Ishii et al., 2016*; *Kiyokawa et al., 2004*, *Kiyokawa et al., 2007*). Hence, the decrease in freezing by mother rats reported here is unlikely to be specific to the presence of offspring. Whether a mechanism for self-defense suppression has evolved in the context of parental defense, which was then co-opted for other forms of social interactions, or the other way around is difficult to assert. Still, this mechanism allows pup protection. Indeed, in our study we show that mother rats do not stay indifferent to the threat, but rather actively direct their behavior towards the threat or the pups.

The present studies also show that oxytocin in the central amygdala underlies the suppression of maternal freezing required for active defense of pups. Specifically, we found that rat dams injected with the oxytocin antagonist, OTA, in the CeL failed to suppress freezing, displaying robust levels of this behavior despite the presence of their pups. The central amygdala has been proposed to act as switch from freezing to other defense responses (*Gozzi et al., 2010*). In addition, a series of elegant studies describe in detail the mechanisms that could constitute such a switch, focusing on the ability of OT to specifically suppress freezing, without affecting other outputs of the medial sub-nucleus of central amygdala (*Knobloch et al., 2012*; *Viviani et al., 2011*). The present findings show that this switch is required for rat dams to defend their pups when both dams and pups are threatened, revealing a natural circumstance for OT's modulation of CeA function. Maternal defense of pups has been widely studied using the male intruder paradigm. In this paradigm, rat dams actively defend their pups from male intruders that pose a threat to the pups but not the mothers. Oxytocin is released upon the introduction of the male acting on several brain areas thereby modulating maternal aggression (*Bosch, 2013*). Manipulating OT in the central amygdala during maternal aggression has yielded mixed results. Antagonizing OT with OTA microinjection into CeA was shown to increase maternal aggression (*Lubin et al., 2003*). However, in rat dams selectively bread for high anxiety, OTA infusion into the CeA has the opposite effect, i.e. it reduces maternal aggression (*Bosch et al., 2005*). Together with the present report, these findings raise the possibility that oxytocin in central amygdala comes online when the mothers perceive a threat to themselves. Oxytocin in this nucleus would not necessarily drive maternal defensive behavior, but rather work as a gate, inhibiting self-defensive behaviors such as freezing and allowing expression of maternal defenses such as aggression towards the threat.

Lastly, the present study also provides evidence that self-defense suppression by rat dams allows for mother-pup interactions that are crucial for pups to learn about threats, since pups that were exposed to the odor with OTA-injected dams failed to learn to freeze to it. Consistent with previous

studies showing maternal influence on the behavior of the pups (*Perry et al., 2016*; *Sullivan and Perry, 2015*), including learning about threats (*Debiec and Sullivan, 2014*), we found a maternal influence on future defensive behavior of pups. Importantly, in this study, by manipulating oxytocin signaling in the central amygdala of the mothers, we show that the type of defensive response displayed by the mother determines whether fear of the odor is transmitted to the pups. We hypothesize that pup learning is mediated by exposure to short-range chemical cues from the mother, such as alarm odor, which requires close proximity between the mother and pups for exposure (*Chang and Debiec, 2016*; *Debiec and Sullivan, 2014*). In addition, these findings suggest that pups are not learning to freeze to the threat through behavior copying. It remains to be established whether learning by pups relies on associative learning mechanisms, leading to learned responses rather specific to the cue, or on non-associative mechanisms, leading to a more generalized response.

In conclusion, our findings reveal a mechanism that suppresses self-defense responses allowing mothers to protect their offspring, while providing novel insights into the mechanisms by which pups learn to respond to threats from their mothers. In addition, we believe our paradigm constitutes a great tool for elucidating the neural mechanism of maternal behavior, as it relies on the described mechanism of action of oxytocin released by PVN projection neurons into the central amygdala, grounding the search for relevant inputs and outputs to this circuit.

## Materials and methods

### Animals
All behavioral experiments were performed in accordance with NIH guidelines and approved by the Nathan Kline Institute's Institutional Care and Use Committee (Protocol: AP2015-522). Primiparous dams (Long-Evans) were born and bred at the Nathan Kline Institute in Orangeburg, NY. Before use in our behavioral experiments, females were pair-housed in a temperature (20°C) and light controlled (12:12 hr light-dark cycle) colony with *ad libitum* food and water in the main laboratory holding room. Breeding occurred in a separate holding room, two females to one male. Females remained with the males for two weeks, after which they were separated and given an individual home-cage until gestation. Dams were carefully monitored, and upon gestation, litters were culled to 12 pups, six females and six males. At the time of the experiments, pups were in one of two age groups, PN4-6 or PN19-21.

### Behavioral assay
#### Maternal defense experiment
##### Day 1: Habituation
The behavioral assay was conducted at a different location from the vivarium. Dams and their pups were transported, via route one of two routes, to the behavioral room location in their home cages covered with special cage covers to minimize anxiety. Dams and her pups were then placed into behavior observation tanks, and habituated to the behavior room for 15 min. These tanks consisted of a glass rectangle box with a removable lid and 5 cm of shavings (identical to home-cage), enough for nest building and general comfort.

##### Day 2: Fear conditioning
Dams and their pups were again transported to the behavior room, via the second route. Alone, dams were conditioned to fear an odor (peppermint, McCormick). A single trial consisted of one 30 s exposure to the peppermint odor (2 L/min flow rate at a concentration of 1:10 peppermint vapor; olfactometer controlled by ChronTrol, ChronTrol Corporation, San Diego, California), which co-terminated with a single 0.7 mA, 0.5 s shock. After three odor-shock pairings (ITI: 5 min, 1 min, 7 min), dams remained in the conditioning boxes for ten seconds, at which point they were removed and re-introduced to the home cage and pups. The home cages were then covered and transported back to the vivarium. Fear conditioning sessions were conducted in the dark, and recorded with high-definition infra-red cameras.

### Day 3: Odor test

24 hr after fear conditioning, dams and their pups were transported to the behavior rooms via route one. Dams, either in the presence or absence of six pups, were placed into the behavior observation tank. After 5 min, the peppermint odor was delivered into the behavior observation tank (procedure for odor delivery as described above). After three 30 s odor presentations, (ITI: 5 min (baseline), 3 min, 3 min), dams and pups were re-introduced to the home cage, covered and returned to the vivarium. The odor tests were conducted in the light and recorded with high-definition web cameras from two perspectives.

## Matched and mismatched pup experiment

For this experiment, we used a subset of the dams which served as controls for our study on the role of OT in the suppression of maternal freezing (Figures 3 and 4); intact dams (PP4-6, n=4 and PP19-21, n=5) and PBS-dams (PP4-6, n=6 and PP19-21, n=7). These dams underwent the same experiment protocol in the maternal defense experiment. The matched and mismatched pup experiment protocol proceeded as follows: 24 hr after the odor test (last session of behavioral assay described above), rat dams underwent a re-conditioning session where they received a single, co-terminating, odor and shock delivery (ITI: 5 min). Upon home cage re-entry, rat dams were given a new litter of non-matched pups (i.e. PP4-6 dams were given PN19-21 pups, and PP19-21 dams were given PN4-6 pups). The next day, rat dams were re-exposed to the odor in the behavior chamber with the mismatched litter. Pups used in the matched and mismatched experiment were not used in the maternal transmission of fear experiment. It should be noted that surrogacy is a standard practice in the Sullivan lab, to which neither rat dams nor pups display abnormal behaviors. We examined the maternal behavior in the home cage of both PP4-6 and PP19-21 dams with both matched and mismatched pups, which when compared were indistinguishable.

## Maternal transmission of fear experiment

PN19-21 pups exposed to the odor in the presence of dams were weaned between PN21-22 and housed together in groups of 6 in new home cages. At PP23-25, the pups were transported in their (covered) home cages to a novel behavior room. Pups were individually habituated to a glass container (25 × 22 cm). After fifteen minutes, the pups were re-introduced to the home cage, covered, and transported back to the vivarium. 24 hr later, the pups were placed back into the behavior assay. After 3 min, a single presentation of the peppermint odor was delivered into the container (delivery procedure as described above). Pups remained in the container for three minutes, after which they were removed and taken back to the vivarium.

## Surgery protocol

Dams were anaesthetized with isoflurane throughout the surgical procedure. Buprenorphine, an analgesic, was injected subcutaneously at the beginning of surgery (0.02 mg/kg). Dams were then placed into a stereotaxic instrument, heads secured with non-puncture ear bars, and sheared. Head position was adjusted so that bregma and lambda were in the same horizontal plane. In aseptic conditions, the skull was prepared and skull screws (Plastics One) were set in place. Using stereotaxic coordinates (derived from Paxinos and Watson, 2007) bilateral craniotomies were performed above the CeM using the following coordinates: 2.2 mm posterior, +/-4.3 mm lateral, and −7.8 mm ventral. The dura matter was then retracted, and guide cannulae (7.5 mm below pedestal) were lowered until 1 mm above the CeM. The cannulae were fixed to the skull with dental acrylic, and secured via the skull screws. Dummy cannulae (1 mm projection) were inserted bilaterally to prevent clogging. Following surgery, dams were kept warm and under observation until recovery from anesthesia and then re-introduced into their home cage and their pups.

## Infusion protocol

Infusions took place on experimental day 3, fifteen minutes before the odor test session. Dams were infused with either 1.0 µg/1 µl of oxytocin receptor antagonist d(CH$_2$)$_5$ [Tyr(Me)$^2$, Thr$^4$, Orn$^8$, des-Gly-NH$_2$$^9$]-vasotocin (OTA) or 0.01M phosphate- buffered 0.9% saline (PBS). The injection cannulae were filled with mineral oil and connected to a 1 µl syringe (Hamilton). They were then loaded with OTA or PBS through backfilling. Rats received either an infusion of PBS or OTA (amygdala 0.3 µl) at

a rate of 0.25 μl/min using an infusion pump (Harvard Apparatus). The OTA or PBS was simultaneously infused bilaterally into the CeM, after which the dummy cannulae were immediately reinserted. For the experiment where we tested habituated, rather than conditioned, dam infusions of OTA or PBS were performed as described above.

## Behavior and data analysis

Behaviors displayed by dams were grouped into three categories that together encompassed most behaviors observed: immobility, 'towards the threat', and 'towards pups'. Immobility included freezing and lack of locomotor activity. Dam freezing and immobility behaviors were scored blindly and independently in real time, and then again using Freezescan, CleverSys, Reston, Virginia, United States. Freezing was used as the index of conditioned fear. An episode of freezing was defined as a period in which there was total immobility except for the respiratory-related movement. The behaviors classified as 'towards threat' included pushing of the bedding both towards and at the odor ports, investigative rearing, exploration of the odor ports and digging. Behaviors classified as 'towards pup' included nursing, grooming, and active time spent with pups. The scoring of maternal behaviors was conducted using a custom Java applet, described in (*Cain and LeDoux, 2007*), which time stamps the beginning and end of keystroke depression relative to the session start; different keys were used for each behavior, which allowed for scoring of multiple behaviors simultaneously. The final behavioral values are the mean of multiple blind and independent scorings.

## Statistics

All statistical analyses were conducted using MATLAB and Statistics Toolbox Release 2014b, The MathWorks, Inc., Natick, Massachusetts, United States. The normality and homogeneity of variance of the data were tested, and the appropriate statistical analyses were used as required. Depending on the comparison, a non-parametric (either a Mann-Whitney U or a Wilcoxon Signed-Rank) test was performed.

Box plots were used to represent the data and corresponding statistical values. For these plots we used behavioral scores taken during the 180 s following the first odor delivery normalized by a baseline period of the same length (values from the 180 s time period before the first odor delivery were used). Hence, the first two minutes of baseline were not used for quantification of the behavioral scores (normalized score = behavior after odor – baseline behavior). For *Figure 2*, rather than using scores normalized to the baseline, which reflect change in behavior triggered by odor delivery, we used absolute values from the time period after the first odor delivery, as our goal was to compare the behavior during odor exposure of dams with matched and non-matched pups (similar results were obtained when using normalized behavioral scores). The central red line indicates the median value. The bottom and top box edges are 25% and 75% percentiles. The whiskers extend to the most extreme data points, not considered outliers. Outliers (greater than $q3 + 1.5 \times (q3 – q1)$ or less than $q1 – 1.5 \times (q3 – q1)$. $q1$ and $q3$ are the 25th and 75th percentiles of the sample data, respectively) are represented individually using the plus (+) symbol.

To quantify changes in behavior of dams over the time course of the experiment we used 1 s time bins. For the raster plots we used 1 s time bins, such that the if a dam transitioned from one behavior to the next, in the middle of a time bin, the time bin would take up the identity of the behavior that lasted longer within that second. To plot the percent time spent in a particular behavior over the course of the test session we computed a running average using 40 s time bins with a sliding window of 1 s. Freezing behavior of pups, along the time course of testing, was scored in 15 s bins, hence no smoothing was required to plot percent time spent freezing.

Sample sizes used in these experiments (7 to 10 animals) were consistent with those typically used in fear conditioning experiments, and the experience with fear conditioning paradigms of the Moita lab. In addition, two pilot studies with a sample size of 6 rats, testing the behavior of intact dams alone or with their pups and implanted dams infused with vehicle or OTA revealed that this sample size was sufficient to demonstrate significant differences between groups. For all experiments performed, a small number of subjects of each experimental group was tested in tandem, such that to achieve the final sample size each experiment was replicated three times. For the infusion experiments, we ran two control groups, an intact group and a vehicle infusion group. Since no statistical differences were found in the behaviors displayed before and after odor exposure

between these groups, data from these was pooled (Mann-Whitney-U tests comparing immobility, behaviors towards threat and towards pups by vehicle and intact PP4-6 and PP19-21 dams before and after odor presentation, all showed significance values corresponding to p>0.3). For the infusion experiment of PP19-21 dams we used more intact dams (n=10), since a subset of these was used for testing learning by the pups (data not shown).

## Histology

After the conclusion of the experimental protocol, dams were deeply anesthetized using isoflurane. Dams were then decapitated and brains immediately frozen using liquid nitrogen. Brains were then stored separately at −80°C until cut on a cryostat into 40 µm thick coronal sections covering the entire extent of the CeA. Sections were analyzed and photographed confirming that cannulae placement was in specified location (see *Figures 3* and *4*). The behavioral data for rats with either one or both cannulae outside of the target area were excluded.

## Acknowledgements

We would like to thank Maria Cano Colino for her technical assistance in programming and data analysis, Syrina Al-Ain for her support with some of the PN19-23 pup experiments, and Karina Szyba for her assistance with the experimental apparatus. The European Research Council (ERC starting grant 337747 CoCO) and NIH (R37HD083217 and MH091451 to RMS) funded this work. ER was supported by Fundação para a Ciência e a Tecnologia and REP was supported by NIH training grant (T32MH096331).

## Additional information

### Funding

| Funder | Grant reference number | Author |
|---|---|---|
| H2020 European Research Council | 337747-C.o.C.O. | Elizabeth Rickenbacher<br>Marta A Moita |
| Fundação para a Ciência e a Tecnologia | SFRH/51257/2020 | Elizabeth Rickenbacher |
| National Institutes of Health | R37HD083217 | Rosemarie E Perry<br>Regina M Sullivan |
| National Institutes of Health | MH091451 | Rosemarie E Perry<br>Regina M Sullivan |

The funders had no role in study design, data collection and interpretation, or the decision to submit the work for publication.

### Author contributions

ER, Conceptualization, Formal analysis, Investigation, Methodology, Writing—original draft, Writing—review and editing; REP, Conceptualization, Investigation, Methodology, Writing—review and editing; RMS, Conceptualization, Resources, Supervision, Methodology, Writing—review and editing; MAM, Conceptualization, Resources, Formal analysis, Supervision, Funding acquisition, Methodology, Writing—original draft, Writing—review and editing

### Author ORCIDs

Elizabeth Rickenbacher, http://orcid.org/0000-0002-4055-5663
Marta A Moita, http://orcid.org/0000-0002-5144-6905

### Ethics

Animal experimentation: All behavioral experiments were performed in accordance with NIH guidelines and approved by Nathan Kline Institute's Institutional Care and Use Committee (Protocol: AP2015-522).

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
