## [Decision Letter]

Thank you for submitting your article "Freezing Suppression by Oxytocin in Central Amygdala Allows Alternate Defensive Behaviors and Mother-Pup Interactions" for consideration by *eLife*. Your article has been favorably evaluated by four reviewers, one of whom, Peggy Mason (Senior Editor), is a member of our Board of Reviewing Editors. The following individual involved in review of your submission has agreed to reveal their identity: Josh Neunuebel (Reviewer #4).

The reviewers have discussed the reviews with one another and the Reviewing Editor has drafted this decision to help you prepare a revised submission.

Summary:

This is an exciting piece of work that shows modulation of fear immobility by the presence of pups. The phenomenon is robust with dams showing freezing in the absence of pups, orientation toward the conditioned fear stimulus or toward the pups in the presence of young and old pups, respectively. The age-dependent pup effect is dependent on mom's stage initially and then on pup's stage. Injection of an OXT-R antagonist into Ce-Amygdala blocks the pups' modulation of mom's fear response, so that the moms freeze now even with the pups around. Finally, the pups learn to fear the odor not from the moms' freezing but from the moms' modulated reaction (not freezing).

Essential revisions:

There are detailed reviews below that are there to help you prepare your revision. The changes that must be addressed are:

Better justify that the modulated behavior is pup protection *or* soften the conclusions;

Change language to acknowledge that OXT-R involvement is tested for NOT OXT release;

Calculation of the% : what are the bin sizes?

Are the responses really starting before the stimulus at t=0? The lines between pre and post are steep and straight and give this impression. if there is a break in data collection to introduce the stimulus, please illustrate this. In any case, please clarify;

Figure 2 does not match the text;

Neutral stimulus for mom to pup transmission experiment is needed to rule out overall increase in anxiety/fear *or* tone down discussion;

Improve or omit videos.

*Reviewer #1:*

This is an exciting piece of work that shows modulation of fear immobility by the presence of pups. The phenomenon is robust with dams showing freezing in the absence of pups, orientation toward the conditioned fear stimulus or toward the pups in the presence of young and old pups, respectively. The age-dependent pup effect is dependent on mom's stage initially and then on pup's stage (see below for a concern with the reporting of this result). Injection of an OXT-R antagonist into Ce-Amygdala blocks the pups' modulation of mom's fear response, so that the moms freeze now even with the pups around. Finally, the pups learn to fear the odor not from the moms' freezing but from the moms' modulated reaction (not freezing). The discussion of this latter result should be augmented a bit to mention the lack of pup copying of mom's behavior.

Figure 2 does not match the text. As the text reads dams of young pups (matched) orient toward the threat when tested with young pups and yet Figure 2 has them orienting toward the pups. And in 2C, it appears that the dams of older pups (matched) orient toward the threat when the text says that they orient toward the pups. Please explain.

Subsection “Regulation of dams’ defensive behavior by the presence of pups”, first paragraph: doesn't maternal aggression increase between birth and weaning?

Please comment on how the sample sizes were determined. The numbers in the groups are greatly different. E.g. 17 controls and 7 OTA.

The videos appeared inconsistently sized on my browser. Moreover, they are simply not useful as is. Please label the time and location of the smell. Annotate the videos (in the videos) so that a viewer can easily glean the point of each video.

From the videos that I could see, it appears that the dams of older pups interact with them closely through taction and one would guess calls as well as odors. The possibility that either taction or audition plays a role in communicating threat to the pups should be at least mentioned.

*Reviewer #2:*

While the results presented in the manuscript by Rickenbacher et al. are novel and interesting, I have some concerns that should be addressed before the paper is published.

1) I have a concern about the interpretation of the results. It is not clear whether increased social contact of the dams with older pups (PP19-21) is a switch from self-defense to offspring protection. The data do not show if the suppression of freezing is limited to the presence of pups or if this suppression occurs in the presence of any conspecific. It is conceivable that the dams engage in social contact with their pups just as they would do with unrelated adult animals (which could result in social buffering effects). Therefore, the assumption should be better justified, either by an additional control group or by a reference to already available results.

2) The sentence "Next, we tested our hypothesis that pup presence leads to release of oxytocin by PVN neurons projecting to the CeL, which suppress freezing and permits expression of maternal defensive behaviors." is an overstatement. There is no data here directly showing that oxytocin is actually released in the CeA when the CS is presented to the dams with pups. Similarly, the specific projections as well as the specific region of the CeA were not targeted in the presented experiments. Furthermore, the abbreviations for the central amygdala are inconsistent throughout the text. In the Introduction, there are references to the CeL (centro-lateral amygdala), in Materiasl and methods, Surgery Protocol the central amygdala is abbreviated to the CeM (which is a common abbreviation for the medial part of the central amygdala). It should be clearly stated that the extent of infusion site is not verified here – so the authors can say that they target the central amygdala, but not one of its parts. Since it is known from other works that the receptors for oxytocin are in the lateral and capsular subdivisions of the central amygdala (Huber et al., 2005), the results suggest that these parts are crucial for oxytocin effects.

3) In the description of the methods:

A) The dose of the oxytocin antagonist is not given. It is particularly important in the context of mixed results on the role of oxytocin in the CeA in maternal aggression. This issue should be addressed.

B) In the Materials and methods section, the ITIs between CSs during odor test are described as 5min, 3 min and 3 min. In figures all ITIs are 3 min. Moreover, the response to the third stimulus is not shown.

C) It should be clearly stated how the freezing scores shown in the results were obtained. It is not clear as it is described now.

D) "The final behavioral values are the mean of multiple blind and independent scorings." – it should be clearly stated how many.

E) It is not clear for which periods of time (3-min bins?) the δ scores showing change of behavior upon odor presentation are calculated.

4) There is a difference in the amount of time "toward pups" before the presentation of the CS (in the baseline) between dams of younger and older pups (Figure 1); this should be discussed.

5) In the last experiment on transmission of fear from mother to pup, there is some variability in the freezing response. Is there any correlation between freezing and intensity of social contact with the dam during the test? Are there differences between pups tested with different dams?

6) The data shown in Figure 2 are inconsistent with the description in the text, probably graphs should be in different order.

7) Why are different behavioral measures shown in Figure 1 and Figure 2 – δ time and time during odor, respectively?

*Reviewer #3:*

There is a lot to like in this paper. The finding that dams express conditioned fear responses that depend on both the presence of pups and the pups' age is novel and provocative. The identification of oxytocin as a permissive factor for the expression of the defensive response demonstrates a potential mechanism. I also really like the raster plots showing behavior for each animal as well.

One thing that I'm finding unclear is exactly how the data for all of the% time graphs (e.g. Figure 1) were calculated. I don't know if I'm missing something somewhere but what are these values a percentage of? Is it across the entire session? If there is a "binning" of time blocks somewhere I can't find it. Normally in cued conditioning experiments there is a time block (e.g. a 30s tone) and freezing% is calculated by what% of those 30s the animals spend freezing. However, since we have continuous line graphs here across time, this doesn't seem to be the approach. Along the same lines, I don't see where the authors describe how the change in% is calculated. Is it the change from before the first presentation of the odor to some kind of aggregate of the behavior during the odor presentation? More detail is needed in exactly where all of these% values come from (again, unless I'm missing where this information is in the manuscript).

*Reviewer #4:*

Rickenbacher et al. conducted a series of elegant experiments to address how self-defense behaviors are suppressed when switching to offspring protection. The authors reported that dams condition to fear an odor froze when tested alone, but when pups were present, they remained in close contact with them or targeted the threat. Oxytocin (OT) signaling in the centro-lateral amygdala (CLA) appear to be crucial for this behavior because blocking OT by injecting oxytocin antagonous into the (CLA) prevented suppression of maternal freezing. Moreover, pups exposed to the odor in the presence of the conditioned dam later froze when re-exposed alone. However, if oxytocin signaling in the dam had been blocked, pups failed to learn. The authors concluded that OT in the CLA plays a role in suppressing self-defense behaviors and allows mothers to protect their pups and transmit information about danger to their offspring. Although the results are striking and fill a gap in knowledge about the circuitry underlying maternal behaviors during innate behavior, a component that needs to be emphasized more in the paper, there are three major, yet addressable concerns. One of which, requires additional experiments.

First, the reported behavioral changes of the dams in Figure 1–Figure 4 all start before the delivery of the odor. The timing of this response is odd because behavioral responses to odor delivery typically occur with a slight delay and not before the odor is delivered. There is clearly a difference in the behaviors of the dams with pups and without pups. The question is what are they mothers really responding to the environment. It doesn't appear to be the odor.

Second, the text for Figure 2 does not match data in the figure. For Figure 2, shouldn't PP4-6 matched replicate your results in Figure 1? Shouldn't PP19-21 matched replicate your results in Figure 1? This is not the case. For Figure 2, it looks like the plots were mislabel and this is concerning. Therefore, the raw data needs to be shown like Figure 1.

Third, you claim that the mothers are transmitting the learned fear response to the pups, but you haven't rule out that the pups could just be over anxious and more sensitive to any stimulus. A vital control was omitted. You needed to also show that animals are not freezing to a neutral odor. If the pups from the control dams don't show freezing to a neutral odor, then you say that they learned the association between peppermint and threat from the mom and make the claim that this learning was transmitted from the moms. Without doing this experiment, then you need to tone down the conclusions you make in the Discussion.

---

## [Author Response]

*Essential revisions:*

*There are detailed reviews below that are there to help you prepare your revision. The changes that must be addressed are:*

*Better justify that the modulated behavior is pup protection or soften the conclusions;*

Reviewers expressed concern regarding our interpretation that increased contact with older pups is a form of maternal defense. We agree with the reviewer in that our experiments cannot establish whether the goal of the observed maternal behavior is to protect the pups. Still, we believe that whether this was the goal or not, increasing contact with the pups’ results in protection by preventing pups from approaching the threat. Since we cannot prove that this is the case we have changed our conclusions to read:

“Conversely, rat dams with older and mobile pups, display increased contact with them. One possibility is that increased contact with the pups, which keeps them in close proximity, constitutes a form of defense by preventing the pups from approaching the threat”.

Furthermore, we added a section in the Discussion about social buffering between adult conspecifics: “Previous studies have shown social buffering of anxiety and fear in rodents (Burkett, Andari, Johnson, Curry, de Waal, & Young, 2016; Hennessy, Kaiser, & Sachser, 2009; Kikusui, Winslow, & Mori, 2006; Waldherr & Neumann, 2007). […] Indeed, in our study we show that mother rats do not stay indifferent to the threat, but rather actively direct their behavior towards the threat or the pups.”.

*Change language to acknowledge that OXT-R involvement is tested for NOT OXT release;*

Reviewers point out that our experiment does not assess whether OT is released by PVN neurons upon the presence of pups. We thank the reviewer for requesting clarification. We have changed the text to read: “Next, we tested our hypothesis that activation of OT-receptors in CeL suppresses freezing, permitting expression of maternal defensive behaviors.”

*Calculation of the% : what are the bin sizes?*

The reviewers asked for clarification regarding the bin size used to calculate the percent time spent in a particular behavior. We score the behavior of dams using annotation software that allows recording the start and end of any behavior with sub-second precision. To create the raster plots we used 1s time bins, such that the if a dam transitioned from one behavior to the next, for example from nursing to exploration, in the middle of a time bin, the time bin would take up the identity of the behavior that lasted longer within that second. Once we had the behavior in 1s time bins we calculated percent time spent in a given behavior using a window of 40 time bins, i.e. of 40 seconds, for the time course plots (see answer below) and for the box plots we calculate percent time spent in a particular behavior using 180 time bins, i.e. 180s. To calculate freezing by pups we used a bin size of 15s for the time course plot and used scores from the three-minute baseline and three minute post first odor exposure (i.e. 12 time bins each) for the box plots. This is now clearly stated in the Methods, statistics section (second and third paragraphs).

*Are the responses really starting before the stimulus at t=0? The lines between pre and post are steep and straight and give this impression. if there is a break in data collection to introduce the stimulus, please illustrate this. In any case, please clarify;*

The reviewers point out that the responses observed seem to start before the onset of the first odor presentation. This results from the use of a smoothing algorithm for the time course plots: running average using 40s time bins with a sliding window of 1s. We decided to keep the plots with a smoothed average (except for the freezing behavior of pups), since we show the raw data in the raster plots, and added a clarification in the Methods, statistics section (third paragraph).

*Figure 2 does not match the text;*

The reviewers report that there is a mismatch between the text and Figure 2. We thank the reviewers for point this out as it results from a mistake in the figure, where we inadvertently had the graphs for behaviors towards threats and pups swapped. We have corrected Figure 2.

*Neutral stimulus for mom to pup transmission experiment is needed to rule out overall increase in anxiety/fear or tone down discussion;*

The reviewers ask to address the specificity of learning by pups with an experiment, or tone down our conclusions. Although we did not make any statement regarding specificity or associativity of learning by the pups, to make sure that readers do not infer from manuscript such a conclusion, we have now added a section in the Discussion addressing this issue: “It remains to be established whether learning by pups relies on associative learning mechanisms, leading to learned responses rather specific to the cue, or on non-associative mechanisms, leading to a more generalized response.”

Improve or omit videos.

The reviewers ask to improve, by annotating, or remove the videos. We decided to make a single video showing the behavior of intact mothers alone or in the presence of young and old pups. This video is clearly annotated to convey the experimental condition of the rat dams, and time point within the experiment. As all four experimental conditions are represented in a single frame, we feel that a direct comparison of the differences in maternal behaviors during odor onset is much easier to observe.

In addition to the essential revisions requested, we have addressed other issues raised by the reviewers, which we thought improved the manuscript, and implement grammar and wording corrections suggested.

1) We added at the end of the Methods, statistics section, a justification for the bigger sample size for control dams in the infusion experiment with PP19-21 dams. We used an additional set of intact dams in this experiment for testing the behavior of pups. Since these dams were tested in tandem with the others we decided to show all the data acquired from all dams tested.

2) We altered Figure 3 and Figure 4 so that the raster plots clearly indicate the behavior of intact and vehicle infused dams. In addition, we added in the Methods, statistics section, a sentence stating the tests comparing intact and vehicle infused dams, which revealed no significant differences: “For the infusion experiments we ran two control groups, an intact group and a vehicle infusion group. Since no statistical differences were found in the behaviors displayed before and after odor exposure between these groups, data from these was pooled (Mann-Whitney-U tests comparing immobility, behaviors towards threat and towards pups by vehicle and intact PP4-6 and PP19-21 dams before and after odor presentation, all showed significance values corresponded to p>0.3).”

3) We added information on the dosage used of OTA to the Methods, in the section regarding the infusion protocol (1.0µg/1µl).

4) We clarified that we targeted the whole CeA, but that OT receptors are expressed only in CeL, so effectively that is the region that our infusions affect: “To this end, we implanted dams bilaterally with injection cannulas into central amygdala (although we did not specifically target CeL, only this region of central amygdala expresses OT-receptors, (Huber, Veinante, & Stoop, 2005)”.

5) We added a sentence in the Results, section on learning from mothers about threats, addressing the variability in the levels of freezing displayed by the pups. The variability was similar within and between litters: “Given the variance in the levels of freezing displayed by pups of CONT-dams, we investigated whether these were more similar within than between litters and found similar variance within and between litters (data not shown).”

6) We added a sentence in the Discussion mentioning that our results suggest that in our experiments pups do not seem to be learning through behavior copying: (Discussion, third paragraph).